# Message framing and COVID-19 vaccine acceptance among millennials in South India

**Aslesha Prakash[1], Robert Jeyakumar Nathan[2]\*, Sannidhi Kini[3], Vijay Victor[3,4]**

**1** Department of Psychological and Behavioural Science, London School of Economics and Political Science Queens House, London, United Kingdom, **2** Faculty of Business, Multimedia University, Melaka, Malaysia, **3** CHRIST (Deemed to be University), Bangalore, India, **4** College of Business and Economics, University of Johannesburg, Johannesburg, South Africa

\* robert.jeyakumar@mmu.edu.my

**Data Availability Statement:** The data underlying the results are available at this link: https://github.com/vjvictor/Vaccine-Hesitancy_Data.

**Funding:** The Article publication charges (APC) for this paper is funded by Multimedia University,

## Abstract

Vaccine hesitancy and refusal remain a major concern for healthcare professionals and policymakers. Hence, it is necessary to ascertain the underlying factors that promote or hinder the uptake of vaccines. Authorities and policy makers are experimenting with vaccine promotion messages to communities using loss and gain-framed messages. However, the effectiveness of message framing in influencing the intention to be vaccinated is unclear. Based on the Theory of Planned Behaviour (TPB), this study analysed the impact of individual attitude towards COVID-19 vaccination, direct and indirect social norms, perceived behavioural control and perceived threat towards South Indian millennials' intention to get vaccinated. The study also assessed the effect of framing vaccine communication messages with gain and loss framing. Data was collected from 228 Millennials from South India during the COVID-19 pandemic from September to October 2021 and analysed using PLS path modelling and Necessary Condition Analysis (NCA). The findings reveal that attitudes towards vaccination, perceived threat and indirect social norms positively impact millennials' intention to take up vaccines in both message frames. Further, independent sample t-test between the framing groups indicate that negative (loss framed message) leads to higher vaccination intention compared to positive (gain framed message). A loss-framed message is thus recommended for message framing to promote vaccine uptake among millennials. These findings provide useful information in understanding the impact of message framing on behavioural intentions, especially in the context of vaccine uptake intentions of Millennials in South India.

## 1. Introduction

The COVID-19 pandemic posed a major threat to the health systems of all countries across the globe. Despite the introduction of vaccines which have significantly reduced severe COVID-19 symptoms and death, vaccine hesitancy and refusal by the public is a barrier in India to improving global public health. According to WHO, vaccine hesitancy is characterised by delay in accepting, hesitation, or denial of vaccines in the face of available immunisation

Malaysia. The funders had no role in study design, data collection and analysis, decision to publish, or preparation of the manuscript.

**Competing interests:** The authors have declared that no competing interests exist.

services and it is just as prevalent in India as in the rest of the world. Various studies in India [1–3] revealed that vaccine safety, rumours and controversies in terms of the adverse effects following immunisation; inadequate knowledge about vaccine benefits, costs and traditional cultural beliefs have been cited as major reasons for vaccine hesitancy in the domain of child-hood immunisation. Moreover, with false information and fake news surrounding COVID-19 being on the rise especially on social media [4] COVID-19 vaccine hesitancy could increase amongst the millennials.

As observed, vaccine acceptance is quite complex and specific to the context, based on the geographic location, culture and behavioural nature of the society in question [5]. It can also be debated that the level of severity and the extent to which the pandemic is likely to affect the individual would serve as a good predictor to assess their intention to get vaccinated [5, 6]. The willingness to vaccinate may in turn depend on how the behaviour recommendations and health consequences are framed in a message and this is a factor that needs to be addressed when communicating health information [7]. What remains a challenge to date is to find an intervention that has the capability to entail effective communications about the vaccines. While a few studies focus on the impact of message framing during the COVID-19 pandemic [8, 9], the lack of literature in the Indian context makes this study unique and a necessity.

The Ministry of Health and Family welfare in India identified four critical areas to be targeted as part of the communication strategy for COVID-19 vaccination; vaccine introduction, hesitancy, eagerness and following COVID-19 appropriate behaviours. The aim was to identify messages and key messengers to disseminate information about the safety and efficacy of the vaccines to not just the traditionally resistant/hesitant groups but also to the general public at different levels to build trust [10]. However, despite the holistic approach on communication strategies, COVID-19 vaccine hesitancy prevails in India. In fact, a study that reported the findings of an online survey called the "Covid Symptom Survey" among Facebook users revealed that a significant portion of the participants (29%) were hesitant to get vaccinated and the reason was mainly attributed to the safety of the vaccines [11]. This necessitates the need to understand how health communications can be effectively used as part of interventions to mitigate vaccine hesitancy.

Persuasive messages are often seen to have the ability to empower individuals to follow healthier lifestyles or change negative ones, according to Public health experts [12]. Goal framing theory, which is often used in persuasive communication research, states that gain-framed messages emphasise the benefits of executing an action, while loss-framed messages emphasise the disadvantages of not exhibiting a particular behaviour [13] as cited in [7]. Along the same lines, the work on Prospect Theory reiterates that substantially identical information may have significantly different effects on people's decisions based on how it is framed [12, 14]. For instance, two completely identical messages with respect to their subject matter differ only on how they have been framed (one that highlights the positive outcome while the other the negative outcome) is perceived differently. This implies that effective gain-framed messages emphasise the advantages of taking a risk, loss-framed messages, on the other hand, emphasise the costs of not taking a risk. Further research is required on how to better develop and convey vaccine related messages to the public, considering not all communication techniques are successful in promoting vaccination intention and some may even turn out to be counterproductive [7].

It was observed in previous studies that young adults are more likely to perceive vaccinations as harmful and unsafe than older adults [15, 16]. Moreover, millennials largely turn to social media communication for pandemic related information [17] which may shape negative views or misinformation towards vaccinations, emphasising the importance of effective persuasion messages.

Since there is a dearth of literature regarding the framing effects of vaccination messages to people during the COVID-19 pandemic, especially in the Indian context; this study seeks to bridge this gap. Moreover, with the pandemic on the rise with multiple waves, understanding vaccination intention and the factors affecting the uptake becomes pivotal. This study therefore, aims to investigate the vaccination intention among millennials guided by the Theory of Planned Behaviour. Additionally, gain framed messages (positive framing) and loss framed messages (negative framing) are incorporated in the study design to assess message persuasiveness in influencing individual's intention to take up vaccination.

## 2. Literature review

### 2.1. Framing effects and vaccination intention

The framing effect describes how people's decisions are influenced by how options are presented or framed and whether potential outcomes are presented in terms of benefits or losses. The framing effect has been studied in a wide range of health behaviours. It can be used to guide people toward health-promoting behaviours [18]. A health message may be presented to highlight the benefits (gain-framed message) of engaging in a behaviour or the drawbacks (loss-framed message) of not partaking in the behaviour. Messages that were presented as gains or benefits in contrast to those framed as losses were considerably more probable of steering individuals towards taking up protective behaviours [9, 19]. This provides a useful insight for framing information regarding vaccination intention, where messages framed in terms of gains could motivate individuals to engage in the behaviour.

However, results of past studies are mixed. In an analysis of 34 studies by [19], 12 showed no major impacts of message framing on vaccination intentions. Some studies revealed a positive effect of loss framed messages contrary to the idea that gain framed messages may be more beneficial. Some other studies highlight that neither of the frames differs in their ability to influence vaccination intentions [19]. With inconclusive results from prior studies, further investigations are needed to probe into these concepts further.

### 2.2. Theoretical framework

To get a comprehensive understanding of health behaviours, such as the protective behaviour of vaccination uptake, the Theory of Planned Behaviour (TPB) has been widely used. According to the theory, intentions to take up vaccination are influenced by attitudes, one's own evaluation of engaging in the behaviour and its consequences; the subjective norms, one's analysis of whether the behaviour would be approved by their close circle; and Perceived Behavioural Control (PBC), one's evaluation of whether they are ready to engage in the behaviour [5]. In a previous study, when assessing women's intention to receive human papillomavirus (HPV) vaccination, it was discovered that attitude, norms, and PBC were key indicators of vaccination intention among women [20]. Similarly [21] found that the three concepts also strongly explained parents' intention to vaccinate their children.

To extend the TPB model to the context of this study, 'Perceived Threat' as an additional variable is added to the research model to assess its impact towards millennials' intention to receive COVID-19 vaccine.

### 2.3. Perceived threat towards COVID-19

The rapid increase in COVID-19 cases has been perceived as a health risk and threat [22]. However, the prolonged nature of the pandemic and lower fatality rate makes people become optimistically biased of their chances of contracting the virus. This would not only lower their

risk perception towards the virus, but also may make them hesitant to take up the vaccination [23].

Moreover, as the mild nature of majority of the cases and the considerable degree of already existing immunity in older age groups becomes clear, initial worries of a moderately serious pandemic fades, just like in the 1968 pandemic [24]. This may be due to the fact that the health consequences are minor, so there is little incentive for people to change their actions, despite government guidelines and media attention. It is therefore essential to understand the perception of risk towards the COVID-19 virus among millennials in India, subsequently to predict their vaccination uptake intention.

### 2.4. Attitude towards COVID-19 vaccine

Another aspect that determines intention is the attitude towards the behaviour. Attitudes pertain to the individual's assessment of the behavior, and whether they find it beneficial or not to undertake such behaviour [25]. According to [25], behaviours associated with outcomes that are viewed as desirable are considered favourable, whereas behaviours that yield undesirable outcomes are negatively evaluated.

With respect to the ongoing pandemic, factors that are more likely to be significant impediments to long-term management of the COVID-19 pandemic are the negative attitudes toward vaccines and apprehensions or refusal to get vaccinated [26]. While evaluating the relationship between attitudes towards the COVID-19 vaccine and each individual's intention to get vaccinated, how much they trusted the safety of vaccines played a major role [8, 26]. In fact, factors such as uncertainty of the vaccine efficacy and the potential unintended side effects have been known to play a role in vaccine hesitancy. Since attitudes towards vaccinations are seen to impact the intention to get vaccinated, there is a critical need for a more complex view of these attitudes and its subsequent impact on the intention to be vaccinated during the COVID-19 pandemic.

### 2.5. Perceived behavioural control towards COVID-19 vaccine

Perceived Behavioural Control (PBC) was incorporated into the first version of what now is called the TPB [27]. PBC accounts for circumstances in which individuals may lack sufficient "volitional control" over specific behaviours [28]. An individual's interpretation of his or her capacity to demonstrate a desired behaviour is known as perceived behavioural control. PBC is a composite of perceived control (e.g., the degree of control one has over getting the vaccine) and self-efficacy (e.g., one's belief in their capabilities to get vaccinated).

PBC is included in the model to account for situations in which people plan to behave in one way but end up changing their behaviour due to a loss of self-efficacy (SE) or influence over that behaviour [29]. PBC is interested in the resources and opportunities that either support or inhibit behavioural success. In HPV vaccinations, PBC was seen to positively affect intention to get vaccinated, and in turn led to the behaviour itself [30]. Hence it is crucial to understand the role of PBC in the context of this study.

### 2.6. Social norms towards COVID-19 vaccine

Subjective norms are the sense of social obligation to practice or abstain from engaging in a certain behaviour [25]. Social norms, according to the TPB has two types; Indirect Norms (Injunctive) and Direct Norms (Descriptive). The former refers to the approval of the people important to the individual who is making the decision to take up the vaccination, while the latter denotes how others' behaviour of taking the vaccination affects the decision maker.

Research highlights that both descriptive and injunctive norms are positively correlated with people taking vaccine [23].

Social influence falls under two categories; direct and indirect. Direct social influence is a consequence of an individual swaying another individual's opinion first hand, often through means of persuasion or manipulation. Indirect social influence exhibits subtle psychological proclivities in which an individual's opinion or conduct is affected due to the information available on the other individual's actions [31, 32]. Therefore, injunctive norms are classified as indirect norms and descriptive norms as direct norms.

If social norms can persuade one to behave in a certain way, it also has the ability to negatively impact one's intent to act and may lead to unintended consequences. With respect to the impact of social norms on intention to get vaccinated, it was seen that perceived behavioural intentions of friends who have taken the HPV vaccine or at least considered it, played a role as a strong predictor [8]. Similar roles of norms were seen in a parent's decision to get their child vaccinated [33]. The literature is unanimous in its assessment of the probability that people will attempt to adapt their actions to their perceptions of the behaviour of others. Thus, it is essential to include subjective norms in the framework of this research.

## 2.7. Framing effects and TPB

The potentials of health messages to significantly impact important health choices and behaviours cannot be overlooked. The importance of coherent and sound information cannot be denied to gain public trust surrounding COVID-19 vaccinations especially in this era of infodemic [4]. Of recent, studies on the effects of health messages are seen to be of immense value to cognitive and decision sciences [34]. Various experimental research studies have been conducted to analyse the role of message framing particularly to influence health behaviours. One such study [35] was undertaken to analyse the intent to take up a flu vaccine, concluded that messages that presented benefits only i.e excluding risk disclosure; determined intent more than messages that present benefits with side effects. [34] hypothesised that gain framed messages (positively framed) increased the indulgence of young adults in preventative sexual health behaviour whereas loss-framed messages (negatively framed) were more effective in promoting STD screening or other health detection behaviours. On the other hand [36] examined the effects of combination of attribute and goal framing on the intention to obtain immunisations, and found that message framing influenced attitudes overall but found no substantial impact on actual behavioural intentions or information seeking.

As seen over the years, message framing has the ability to influence the intention to behave [37] and these intentions form a key focus of the TPB. Social and cognitive variables like attitude, subjective norms and perceived behavioural control affect peoples' intentions. While some researchers concluded that these variables are altered by message framing in such a way that they mediated the effect of framing on intentions [13, 37], other studies posit that these variables only moderated the relationship between framing and intentions [37, 38].

In another study [39] carried out a randomised trial to explore the effects of positive vs negatively framed educational brochures amongst inactive colorectal cancer survivors by offering suggestions for increasing physical activity as a means to reduce the recurrence of cancer. The key constructs of TPB were measured in both the frames. While both frames produced remarkable increase in physical activity among the people, there were no significant difference among the efficacy of the frames.

The inability of past literature to produce conclusive results on the debate between loss vs gain framed messages made it necessary to probe into the role of these social cognitive

constructs in relation to message framing and intention to behave in the context of the COVID-19 vaccine uptake.

Based on the discourse above, this study forwards the following hypotheses, from H1 to H7, for measurement and testing.

Hypothesis 1 (H1). Attitudes play a significant role in predicting the intention to get vaccinated.

Hypothesis 2 (H2). Direct social norms play a significant role in predicting the intention to get vaccinated.

Hypothesis 3 (H3). Indirect social norms play a significant role in predicting the intention to get vaccinated.

Hypothesis 4 (H4). Perceived behavioural control plays a significant role in predicting the intention to get vaccinated.

Hypothesis 5 (H5). Perceived threat of contracting COVID-19 plays a significant role in predicting the intention to get vaccinated.

Hypothesis 6 (H6): There is a significant difference of vaccine uptake intention in positive and negatively framed messages.

The research variables and their corresponding relationship are shown in Fig 1.

## 3. Materials and methods

To understand message framing effects on vaccination up-take, hypothetical scenarios were constructed using Positively Framed and Negatively Framed messages. The hypothetical scenarios were built based on available facts, and presented information about the availability of vaccines, their efficacy, the immediate possible side effects and aspect of social norms. The messages were framed based on the information available on the Centre for Disease Control website at the Ministry of Health and Family Welfare, Government of India [10]. The framed messages are presented in Table 1.

Positively Framed messages highlighted the gains and benefits of getting the vaccination, while Negatively Framed messages suggested the losses and potential harms of not being vaccinated. The constructs of TPB namely, Attitudes, Direct and Indirect Social norms, PBC and the additional variable, Perceived Threat were also included through the message framing. The questions that followed measured the constructs of the TPB in order to understand the sample respondents' intention to vaccinate. All items of the research variables were measured using a

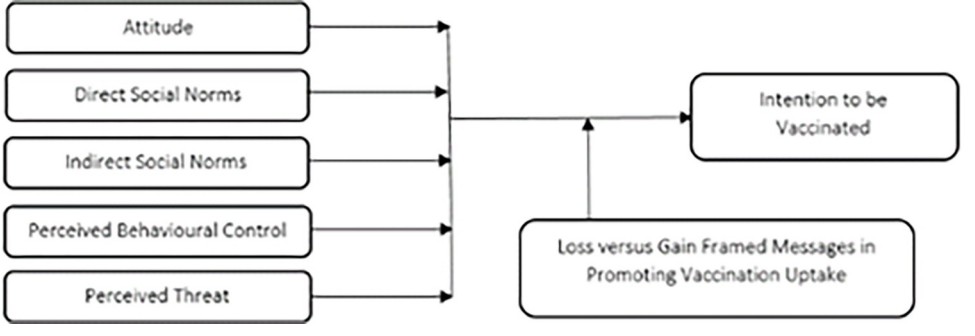

**Fig 1. Research framework.**

**Table 1. Scenarios of positive and negative frames.**

| Frames | Message Content |
|---|---|
| Positive Frame (Ministry of Health and Family Welfare [10]) | Ramesh is a 25-year-old living in the city of Bangalore as an IT professional. He hears news about the COVID-19 vaccination being given to the people of his age group. His family, friends and coworkers feel positively about the vaccination. |
| | While considering whether or not he should take up the vaccination, he reads an article by the Centre for Disease Control and Prevention (CDC) which reads "did you know getting yourself vaccinated will decrease your chances of contracting the virus?". |
| | The vaccinations being given in India demonstrate a remarkable 80% effectiveness. The side effects are pain at the injection site, fever, fatigue and body aches in some cases. However, the benefits of getting vaccinated against COVID-19 far outweigh the risks. It is on Ramesh to choose wisely. |
| | Moreover, if Ramesh chooses to vaccinate himself, he will be able to save himself and his family from contracting the virus. He will also feel less anxious and be able to experience the safety that comes with being vaccinated. |
| Negative Frame (Ministry of Health and Family Welfare [10]) | Ramesh is a 25-year-old living in the city of Bangalore as an IT professional. He hears news about the COVID-19 vaccination being given to the people of his age group. His family, friends and co-workers feel positively about the vaccination. |
| | While considering whether or not he should take up the vaccination, he reads an article by the Centre for Disease Control and Prevention (CDC) which reads "did you know not getting yourself vaccinated will increase your chances of contracting the virus?". |
| | The vaccinations being given in India are seen to not be effective in a mere 20% of the situations. The side effects are pain at the injection site, fever, fatigue and body aches in some cases. However, if he is given a choice to protect himself, his family and his community from the highly transmissible and deadly coronavirus that results in long term health consequences for a large number of otherwise healthy people; it may cost him a few days of feeling sick. It is on him to choose wisely. |
| | Moreover, if Ramesh chooses to not vaccinate himself, he will fail to save himself and his family from the virus. He will also be more anxious and will not be able to benefit from the peace of mind after getting vaccinated. |

five-point Likert scale, where (1) Strongly Disagree, (2) Disagree, (3) Neutral, (4) Agree, and (5) Strongly Agree, that would determine the extent to which the respondents agree to disagree to the particular item. The indicators of this study are adopted from previous sources [40] and [41]. The constructs and the respective items with descriptive statistics are presented in Table 2.

Judgmental sampling technique was employed to recruit participants who were not vaccinated at the time of the data collection of this study. Google forms were distributed randomly among 350 millennials who were mostly university students. The questionnaire started with a question of the vaccination status of the participant and those who have received any dose of vaccination were not invited to participate in the study, only those who have not yet been vaccinated are screened to participate in the study.

The Indian government approved vaccination for those aged 18 years and above on 1st May 2021. At the time of the study, the indigenously made vaccine named COVAXIN was pending to get approval by the WHO. The other vaccine available was the ChAdOx1-S, manufactured by the Serum Institute of India which was given approval for emergency use. Participants were informed that their participation is voluntary and consent was informed in the google survey form. Prior ethics approval for the research instrument under the research

**Table 2. Research variables, average item mean for both frames.**

| Variable | Indicators | Mean | SD |
|---|---|---|---|
| Attitude [40] | A1- COVID-19 vaccine would be beneficial for me. | 4.276 | 0.799 |
| | A2- COVID-19 vaccine would be beneficial for children | 3.89 | 0.965 |
| | A3- COVID-19 vaccine would be beneficial for individuals 60-years and older | 4.491 | 0.775 |
| | A4- COVID-19 is a serious pandemic | 4.399 | 0.885 |
| | A5-COVID-19 vaccine would be beneficial for the health of my community | 4.513 | 0.71 |
| | A6-COVID-19 vaccine is safe | 3.93 | 0.92 |
| | A7- COVID-19 vaccine is effective in preventing COVID-19 | 3.969 | 0.84 |
| | A8- COVID-19 vaccine should be mandatory for all | 3.842 | 1.222 |
| Direct Social Norms [40] | DSN1- Most people who are important to me would think that I should receive the COVID-19 vaccine | 4.25 | 0.9 |
| | DSN2-People who are important to me would expect me to receive the COVID-19 vaccine | 4.241 | 0.912 |
| | DSN3-I would feel under social pressure to receive a COVID-19 vaccine | 3.009 | 1.218 |
| | DSN4-Everyone I know would get the COVID-19 vaccine | 3.724 | 1.021 |
| Indirect Social Norms [40] | ISN1-My family physician (or other primary Health Care Provider) would approve of me receiving a COVID-19 vaccine | 4.373 | 0.809 |
| | ISN2-My family physician (or other primary Health Care Provider) would approve of me receiving a COVID-19 vaccine | 4.202 | 0.86 |
| | ISN3-My co-workers would approve of me receiving the COVID-19 vaccine | 4.224 | 0.837 |
| | ISN4-What my coworkers think is important to me | 3.351 | 1.207 |
| | ISN5-My friends would approve of me receiving the COVID-19 vaccine | 4.289 | 0.845 |
| | ISN6-What my friends think is important to me | 3.702 | 1.096 |
| | ISN7-My family would approve of me receiving the COVID-19 vaccine | 4.36 | 0.839 |
| | ISN8-What my family thinks is important to me | 4.311 | 0.939 |
| Perceived Behavioural Control [40] | PBC1-I could easily receive a COVID-19 vaccine if I wanted to | 3.662 | 1.13 |
| | PBC2-It would be completely up to me whether I received the COVID-19 vaccine | 4.219 | 0.985 |
| | PBC3-I have high control to receive COVID-19 vaccine. | 3.934 | 0.955 |
| Perceived Threat [41] | PT1-I am afraid of contracting coronavirus. | 3.693 | 1.01 |
| | PT2-Coronavirus poses a large personal threat to me | 3.697 | 1.018 |
| | PT3-Coronavirus poses a large societal threat to my community | 4.421 | 0.7 |
| | PT4-I am afraid for my community of contracting and spreading the coronavirus | 4.215 | 0.785 |
| Intention to be Vaccinated [40] | INT1-I am likely to be vaccinated for COVID-19 when a vaccine becomes available | 4.158 | 0.965 |
| | INT2-I would consider vaccinating myself and my family when a vaccine is available to the public. | 4.364 | 0.845 |
| | INT3-I would have already taken the vaccine if it were available. | 4.009 | 1.112 |

project was obtained following the Internal Review Process (IRB) of Multimedia University (MMU) Malaysia, from the Research Ethics Committee (REC) at the MMU Technology Transfer Office (TTO). The approval code for the IRB is EA2992021 (Ref: TTO/REC/EA/299/2021).

The data collection was carried out between August-September 2021. Two manipulation check items were included in both questionnaires to ensure that the respondents understood the frames properly. Inappropriate responses to the manipulation check questions were not included in the final data used for analysis. Three responses that did not pass the manipulation checks were eliminated in the final sample. The final sample had 124 responses in the Gain Frame Scenario, and 104 in the Loss Frame Scenario. This sample number is adequate for data analysis based on G*Power software sample size determination for PLS SEM path modelling. With 5 predictors, 0.05 α error probability and an estimated effect size of 0.15, the sample size of 124 suffices to produce statistical power above 0.90. The sample size of 104 in the negative

frame model would give actual power above 80 with the same parameters. Additionally, the sample size for both frames are consistent with the 10 times rule used in PLS SEM sample estimation which suggest that the sample size should be greater than 10 times the maximum number of inner or outer model links pointing at any latent variable in the model [42].

The responses were analyzed using IBM Statistical Package for Social Sciences (SPSS) for descriptive analysis. Path modelling and Hypothesis testing were done using the Partial Least Square (PLS) Structural Equation Modelling (SEM) on Smart PLS software version 3.0.

The PLS-SEM analysis is suitable when prediction is emphasized over theory testing and when it is difficult to meet the requirements for large samples or identification in SEM [43]. For a study that is driven by predictive modelling as well as by the aim to test relationships between new constructs in the context of COVID-19 vaccination uptake, PLS-SEM seemed to be an appropriate data analysis tool.

Further, the PLS-SEM analysis was chosen with an aim to identify the key factors that have high or low influence on the intention to get vaccinated when the COVID-19 vaccine becomes available rather than just discarding indicators that predict the vaccine uptake to reach the required level of goodness of fit for the model. PLS-SEM analysis is deemed fit given its ability to test new indicators and to assess the model for its overall reliability and validity [43, 44].

## 4. Results

A total of 228 usable responses were collected for both frames. The respondents' demographic details are presented in Table 3. Of the 124 respondents in the Positive Frame scenario, 75% were females, and 49% were males. As for ages, the majority of the respondents were between the ages of 18–25 (88.7%), followed by 6.4% between the ages of 26–35 and 4.9% between 36–45. Likewise, in the Negative Frame scenario, 56.7% were females, with 98% in the 18–25 age group.

The measurement model was evaluated using construct validity, convergent validity and discriminant validity analyses. Prior to performing the hypothesis testing, it was imperative to assess the indicators' factor loadings. Indicators that had loadings below 0.50 were removed from the path model as it indicated low predictability of the relevant variable [44]. Thus, A3, A4, DSN 3, ISN 2, ISN 4, ISN 6, ISN 8 and PBC1 were removed from both the Positive Frame and Negative Frame path models so as to make similar comparisons of path modelling for both the scenarios (refer to Table 4).

All factor loadings were above the threshold level of 0.70 [44] assuring the convergent validity of the constructs. With respect to the Cronbach Alphas, all values were above 0.70, fitting well within the satisfactory values [45]. The values of all constructs and indicators are presented in Table 4.

Variables with high inter-relationship and multicollinearity between them lead to faulty findings, magnified standard errors or weaker power of regression coefficients. If the values of

**Table 3. Respondents' demographic information.**

| Demographic Characteristics | Options | Gain Frame | | Loss Frame | |
|---|---|---|---|---|---|
| | | Freq. | Percentage (%) | Freq. | Percentage (%) |
| Gender | Male | 49 | 39.5 | 45 | 43.3 |
| | Female | 75 | 60.5 | 59 | 56.7 |
| Age | 18–25 | 110 | 88.7 | 102 | 98 |
| | 26–35 | 8 | 6.4 | 1 | 1 |
| | 36–45 | 6 | 4.9 | 1 | 1 |
| TOTAL (N) | | 124 | | 104 | |

**Table 4. Internal consistency, composite reliability and convergent validity.**

| Variable | Indicator | Factor Loadings | Cronbach's Alpha | Composite Reliability | AVE |
|---|---|---|---|---|---|
| Attitude | A1 | 0.802 | 0.872 | 0.903 | 0.609 |
| | A2 | 0.758 | | | |
| | A5 | 0.797 | | | |
| | A6 | 0.810 | | | |
| | A7 | 0.795 | | | |
| | A8 | 0.718 | | | |
| Direct Social Norms | DSM1 | 0.909 | 0.790 | 0.880 | 0.712 |
| | DSM2 | 0.915 | | | |
| | DSM4 | 0.688 | | | |
| Indirect Social Norms | ISM1 | 0.843 | 0.877 | 0.915 | 0.730 |
| | ISM3 | 0.824 | | | |
| | ISM5 | 0.879 | | | |
| | ISM7 | 0.872 | | | |
| Perceived Behavioural Control | PBC2 | 0.858 | 0.752 | 0.887 | 0.798 |
| | PBC3 | 0.927 | | | |
| Perceived Threat | PT1 | 0.729 | 0.798 | 0.865 | 0.616 |
| | PT2 | 0.767 | | | |
| | PT3 | 0.791 | | | |
| | PT4 | 0.849 | | | |
| Intention to be Vaccinated | INT1 | 0.886 | 0.886 | 0.929 | 0.814 |
| | INT2 | 0.921 | | | |
| | INT3 | 0.899 | | | |

Heterotrait-Monotrait Ratio of Correlations (HTMT) are lower than 0.85, it signifies that the variables used are conceptually different from each other [46]. The value of HTMT 0.85 acts as the conservative criterion to assess discriminant validity in the present study.

From Table 5, all HTMT values are below the threshold level of 0.85, indicating that the model ensured adequate discriminant validity.

Further, the Standardised Root Mean Square Residual (SRMR) value was computed in order to gauge the goodness of fit of the research model under study. As indicated by the results, the SRMR value for the merged data set was 0.065. The threshold, as stated by [46] is 0.08 and since the SRMR value was within the given threshold, it assured the goodness of fit of the research models with respect to positive and negative frames. The R square value for the research variable "intention" stood at 0.565. As explained in the path model, the research variables indicated high variance with respect to intention to vaccinate. This implies that the research variables or the model's inputs accounted for approximately 57% of the variance. The R square value for the mediating variable "indirect social norm" was observed to be 0.499.

**Table 5. The Heterotrait-Monotrait ratio of correlations (HTMT).**

| | ATT | DSM | ISM | INT | PBC | PT |
|---|---|---|---|---|---|---|
| ATT | | | | | | |
| DSN | 0.721 | | | | | |
| ISN | 0.761 | 0.821 | | | | |
| INT | 0.781 | 0.605 | 0.723 | | | |
| PBC | 0.280 | 0.262 | 0.354 | 0.247 | | |
| PT | 0.275 | 0.281 | 0.362 | 0.439 | 0.159 | |

**Table 6. Results of hypotheses testing for vaccination intention.**

| Hypothesis | Relationship | Path Coef. | p-Value |
|:---:|:---:|:---:|:---:|
| H1 | A→INT | 0.461 | 0.000 |
| H2 | DSN→INT | 0.001 | 0.991 |
| H2 | ISN→INT | 0.271 | 0.000 |
| H4 | PBC→INT | 0.003 | 0.957 |
| H5 | PT→INT | 0.185 | 0.000 |

## 5. Hypotheses testing

To test the hypotheses, bootstrapping with 5000 iterations was applied. To understand the validity of each hypothesis, the significance of the path coefficient is assessed. The path coefficients from the PLS structural model and p-values are presented in Tables 6 and 7.

Based on the results of the path analysis, it is found that Attitudes (A), Indirect Social Norms (ISN) and Perceived Threat (PT) have significant impact on the intention to take up the vaccination (Hypotheses 1, 3 and 5 are supported). However, the relationship of Direct Social Norm (DSN) and Perceived Behavioural Control (PBC) with intention to take up the vaccination is insignificant (Hypothesis 2, 4 are not supported).

Among the five independent variables (H1 to H5), Attitude (A), Indirect Social Norms (ISN), and PT (perceived threat) have significant impact on the intention to be vaccinated ($p < 0.001$). This indicates the importance of favourable or unfavourable attitudes towards vaccinations that predict an individual's intention to be vaccinated. It also shows the role of indirect social influences while taking the decision to get vaccination and the higher degree of perceived threat towards the pandemic that determines vaccination intention. PBC and ISN, on the other hand, were seen to be insignificant predictors of vaccination intention.

### 5.1. Necessary condition analysis

Necessary Condition Analysis (NCA) is a technique used to identify the essential conditions in a model which cannot be otherwise identified using the traditional methods such as regression and correlation. A necessary condition basically implies that without the right level of a cause variable, the effect won't occur. A necessary condition is a crucial factor of an outcome, and the absence of the condition can result in the failure of the model. However, the presence of a necessary condition does not guarantee the success of a model too; in this case, the condition is necessary but not sufficient. In order to avoid the failure of a model, it is crucial to identify all necessary conditions and the required level of factors in a model [47]. For this study, the independent variables, i.e. attitude, direct social norms, indirect social norms, perceived behavioural control and perceived threat, were tested against the intention to vaccinate using NCA to examine whether they are necessary determinants. Figs 2–6 display the NCA plots for each research variable against Intention.

**Table 7. Effect sizes.**

| Variables | Effect Sizes | | Slope | P Value |
|:---:|:---:|:---:|:---:|:---:|
| | ce_fdh | cr_fdh | | |
| Attitude | 0.134 | 0.123 | 1.208 | 0.000 |
| DSN | 0.060 | 0.047 | 4.422 | 0.013 |
| ISM | 0.307 | 0.291 | 0.969 | 0.000 |
| PBC | 0.048 | 0.034 | 1.215 | 0.286 |
| PT | 0.143 | 0.120 | 2.008 | 0.003 |

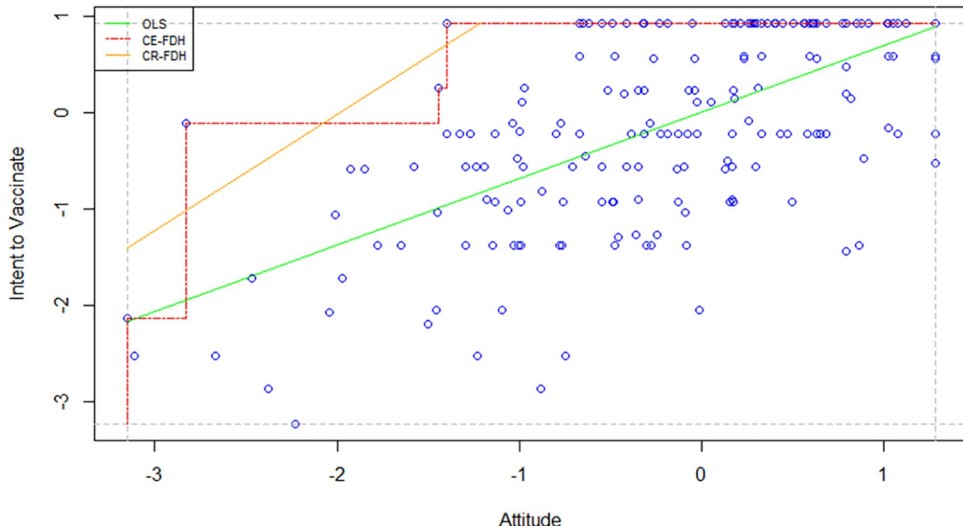

**Fig 2. NCA (attitude—intention).**

The findings from the NCA, including the effect size, the slope of the relationship between the variables, and the significance of the slopes are presented in Table 7.

The effect sizes in Table 7 are statistically significant for all factors except PBC. Indirect social norms seem to have the highest effect size, followed by attitude and perceived threat. The effect size of direct social norms is minimal, although statistically significant. A bottleneck analysis was conducted to further understand the relevance of the research variables to the study. The results are presented in Table 8.

The bottleneck analysis result in Table 8 shows the importance of each necessary condition in the model. The results indicate that 90% of the intention to vaccinate require 60.5% indirect social norms, 35.9% attitude, and 28.5% perceived threat. The share of direct social norms and

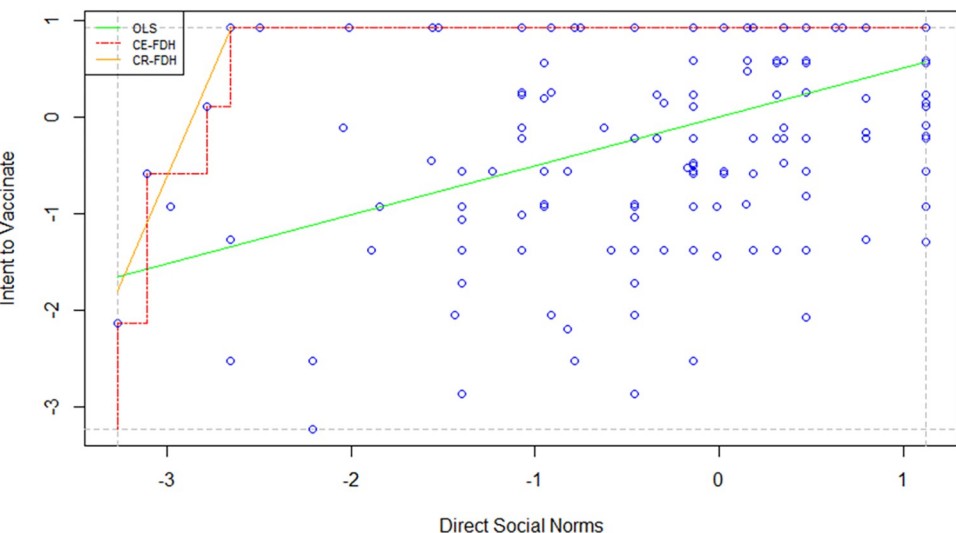

**Fig 3. NCA (direct social norms—intention).**

**NCA Plot : Indirect Social Norms - Intent to Vaccinate**

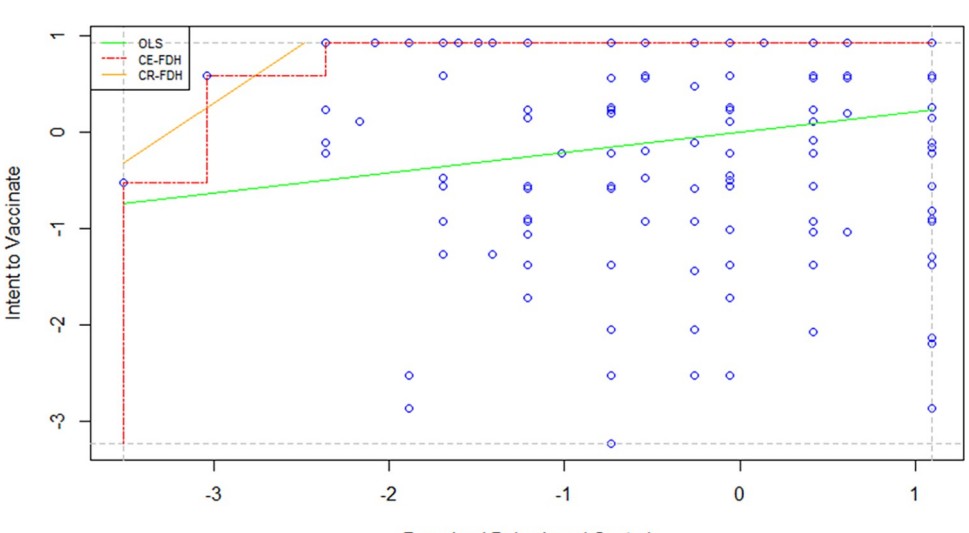

**Fig 4. NCA (indirect social norms—intention).**

perceived behavioural control are 12% and 15%, respectively, implying that these factors don't significantly influence the intention to vaccinate. The results of NCA supplement the PLS SEM results by emphasising the role of indirect social norms, attitude and perceived threat in increasing the intention of millennials to be vaccinated.

## 5.2. Independent sample T test

Independent sample t test is conducted to test for statistical differences between the gain and loss frame scenarios. Table 9 shows the result of the t test, the mean scores and standard deviation for each research variable.

**NCA Plot : Perceived Behavioural Control - Intent to Vaccinate**

**Fig 5. NCA (perceived behavioural control—intention).**

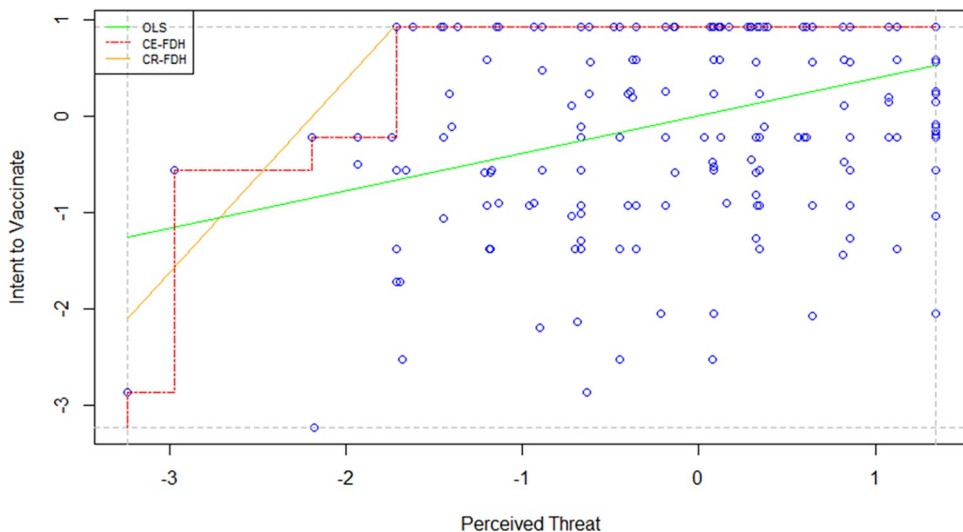

**Fig 6. NCA (perceived threat—intention).**

**Table 8. Bottleneck analysis with ceiling envelopment–free disposal hull.**

| Bottleneck: Intention to Vaccinate | Attitude | DSN | ISM | PBC | PT |
|---|---|---|---|---|---|
| 0 | NN | NN | NN | NN | NN |
| 10 | NN | NN | NN | NN | NN |
| 20 | NN | NN | 3.9 | NN | NN |
| 30 | NN | NN | 11.9 | NN | 1.3 |
| 40 | NN | 1.3 | 20 | NN | 5.8 |
| 50 | 4.9 | 3.4 | 28.1 | NN | 10.4 |
| 60 | 12.6 | 5.6 | 36.2 | NN | 14.9 |
| 70 | 20.4 | 7.7 | 44.3 | 0.2 | 19.4 |
| 80 | 28.2 | 9.9 | 52.4 | 7.6 | 24.0 |
| 90 | 35.9 | 12.0 | 60.5 | 15.0 | 28.5 |
| 100 | 43.7 | 14.2 | 68.6 | 22.05 | 33.0 |

**Table 9. Independent-sample T test.**

| Variable | Gain Frame | | Loss Frame | | F | Sig. | T statistic | p-value (2 tailed) |
|---|---|---|---|---|---|---|---|---|
| | Mean | Std Dev. | Mean | Std Dev. | | | | |
| A | 4.1804 | .59305 | 4.1442 | .61528 | .354 | .552 | 0.451 | 0.652 |
| DSN | 3.7258 | .67882 | 3.9014 | .67138 | .226 | .635 | -1.956 | 0.052* |
| ISN | 4.0544 | .64456 | 4.1575 | .68021 | .214 | .644 | -1.172 | 0.242 |
| PBC | 3.6452 | .61823 | 3.7260 | .58761 | .418 | .519 | -1.005 | 0.316 |
| PT | 3.9476 | .68911 | 4.0769 | .70717 | .003 | .957 | -1.395 | 0.164 |
| INT | 4.0887 | .88254 | 4.2821 | .86993 | .187 | .666 | -1.658 | 0.099* |

*Significant at 0.1 significance level

The independent sample t test result reveal significant differences between the two frames in people's intention to be vaccinated. The two message frames significantly differ in their ability to persuade individuals. With the exception of Attitude, all other variables' mean values for loss-frame scenario are found higher than in gain frame; indicating a better efficacy of loss frame message over gain frame message. This result confirms that negatively framed message (loss frame) is more effective in persuading individuals to be vaccinated; with mean for Intention 4.28 (in loss frame) and 4.08 (in gain frame).

## 6. Discussion

The findings of this study reveal multiple key similarities between positive (gain) and negative (loss) frame scenarios. In terms of the intention to get vaccinated among millennials, attitudes towards the vaccination, indirect social norms, and perceived threat play a significant role. Whereas, the impact of direct social norms and perceived behavioural control were insignificant to impact vaccination intention in both scenarios in general. However, further analysis revealed a significant difference in the efficacy of the two frames, where the negative frame resulted in higher vaccination intention among millennials.

This finding is consistent with the idea of loss aversion, which states that 'losses loom larger than gains'; that is, the pain of losing is much greater than the pleasure of equal gains [14]. In fact, from the finding of [48], loss framed messages are seen to work better in times of uncertainty, especially in the context of disease prevention. Infectious disease outbreaks such as the COVID-19 pandemic cause uncertainty among individuals [49] and therefore loss framed messages seemed to fare better in this new and poorly understood risk context. Any risk situation characterized by unknowability and novelty leads to low levels of acceptance by the public [49]. As a result, a dramatic increase in information seeking behaviours was seen since the COVID-19 outbreak. The second wave, which has been much worse than the first, has led to the increasing uncertainty of death and illness, thus revealing a more pronounced efficacy of negatively framed messages in India. Moreover, the perceived threat and severity of the COVID-19 pandemic could have also led to the loss framed message being more effective in influencing individual behaviour [50].

Increased perceived threat towards the pandemic increases vaccination intention, as found in the results of this study, suggesting that despite the pandemic lockdown fatigue [51], increased perceived threat of the COVID-19 virus is a crucial factor in influencing vaccination uptake. Findings of [52] revealed that perceived risk of the COVID-19 virus to oneself was not closely related to the intention to get oneself vaccinated and instead was driven more by the severity it poses to others. This finding justifies the relatively lesser importance of this construct, since the COVID-19 pandemic seemed to affect older adults more than the millenials, just like in the previous pandemics [24]. Therefore, one's own perceived threat towards the pandemic did not play a crucial role here since the study was conducted on millennials who did not face severe health consequences from the pandemic.

Attitudes towards COVID-19 vaccination were favourable, indicating strong positive belief in the importance of vaccine to end the pandemic; analogous to previous literature on COVID-19 vaccinations in adult populations in the UK [26]. This indicates attitudes and beliefs of an individual highly influence vaccine uptake, regardless of their age bracket and that large numbers of people are driven by their own personal beliefs about the vaccine while considering getting vaccinated. Thus, attitudes towards the benefits of the vaccine for oneself, the community; attitude towards the seriousness of the pandemic as well as effectiveness and safety of the vaccines play a crucial role in vaccine acceptance as also evidenced by [53] in a study in the global south.

Similarly, Indirect Social Norms, including family, coworkers, and friends' approval and the individual's motivation to comply with those norms, were revealed to be a significant predictor of intention. [54] in their study on COVID-19 vaccinations reported a strong link between willingness to vaccinate and the importance given to approval of friends, families, and healthcare professionals, thus establishing a strong social norm. This finding suggests that indirect social norms are deemed more important than general social pressures and norms about vaccination uptake regarding COVID-19 vaccinations. This is in agreement with the finding of [33] and [54] where the studies found the role of the social networks such as spouses or family and friends play a much greater role in pro-vaccination behaviours.

On the other hand, Direct Social Norms do not significantly impact intention to get vaccinated in this study, which was found important previously in the study of vaccinations for HPV. Literature indicates that often people tend to misjudge the extent to which people engage in preventive healthcare behaviours. Since direct social norms in this study focused on the perceptions of others' attitude towards vaccine uptake, it may have failed to have the intended impact to intention [55]. By understanding these norms and how they have the ability to impact health behaviours better, a strengthened preventative strategy can be created while limiting the possibilities of normative misconceptions.

This study finds a non-significant relationship between perceived behavioural control and intention to get vaccinated. This could be attributed to the overwhelming nature of the pandemic, as it has been affecting individuals at varying levels thus reducing their control over getting vaccinated [56]. As argued by [56] in the US as well, the COVID-19 pandemic is being viewed as a societal issue rather than a personal one. Therefore, India known for being a collectivistic society, could have resulted in the individual's low perceived behavioural control. More so, because of the systemic societal issues surrounding rules about vaccinations based on age and the scarcity of vaccines in India at the time of the study. Accessibility to vaccines then could have led to the lower perceived behavioural control [57]

## 7. Conclusion

With the coronavirus cases increasing exponentially, vaccination is one of the major weapons available to help curb the spread of the pandemic further [58]. Studying vaccination intentions in such situations therefore becomes vital. Various studies have analysed the framing effects in the context of the pandemic, providing mixed results [19]. The present study has added to the existing literature on the effectiveness of the frames, by showing that negatively framed messages better persuade vaccination intention for COVID-19 vaccination than positively framed messages. This provides a crucial standpoint for the government and policymakers, as not just effective messages but how the messages are framed too have an impact on vaccination intention. Negative or loss framed messages have a significant advantage over positive or gain framed messages with respect to COVID-19 vaccination intentions.

Furthermore, various studies conducted in the past have shown the significance of direct social norms and perceived behavioural control in getting vaccinated [30]. However, with respect to COVID-19 vaccinations, indirect social norms from family, friends and coworkers play a greater role, possibly due to prevailing distrust in the COVID-19 vaccines. Health care providers and policy makers may focus on increasing public trust in vaccines by conveying messages accurately using indirect social norms and focusing on increasing the perceived behavioural control among the unvaccinated population, specifically Millenials.

## 8. Limitations and future studies

This study focused on the Indian context and hence the results are best suited to South India, whilst the framework can be replicated in other regions. Since the focus of the study was Millennials, data from other generations was not collected for the purpose of this study which did not allow for comparisons of findings between different generations and age groups.

The vaccines available in India during the time of this study's data collection period were not yet given full approval by WHO, which could have increased the level of skepticism among participants. As the study was conducted during the second wave of the COVID-19 pandemic in India, there were difficulties in organising large scale data collection. Future studies could consider a larger sample size including other population groups and regions in India, especially comparing rural and urban communities, to get a better perspective on the state of vaccination intention among the diverse communities in India. Further research could also consider other variables such as trust in vaccines, misinformation surrounding the vaccinations on social media, and the persuasiveness of the message source or agent [59] towards people's vaccination intentions.

## Author Contributions

**Conceptualization:** Aslesha Prakash, Robert Jeyakumar Nathan, Vijay Victor.

**Data curation:** Aslesha Prakash, Robert Jeyakumar Nathan, Sannidhi Kini, Vijay Victor.

**Formal analysis:** Sannidhi Kini, Vijay Victor.

**Investigation:** Vijay Victor.

**Methodology:** Aslesha Prakash, Robert Jeyakumar Nathan, Sannidhi Kini, Vijay Victor.

**Project administration:** Robert Jeyakumar Nathan, Vijay Victor.

**Resources:** Robert Jeyakumar Nathan.

**Software:** Sannidhi Kini, Vijay Victor.

**Supervision:** Robert Jeyakumar Nathan, Vijay Victor.

**Writing – original draft:** Aslesha Prakash, Robert Jeyakumar Nathan, Sannidhi Kini, Vijay Victor.

**Writing – review & editing:** Aslesha Prakash, Robert Jeyakumar Nathan, Sannidhi Kini, Vijay Victor.

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
