## [Decision Letter · Decision Letter 0]

12 Apr 2022

PONE-D-22-05789Message Framing and Covid-19 Vaccine Acceptance among Millennials in South IndiaPLOS ONE

Dear Dr. Jeyakumar Nathan,

Thank you for submitting your manuscript to PLOS ONE. After careful consideration, we feel that it has merit but does not fully meet PLOS ONE’s publication criteria as it currently stands. Therefore, we invite you to submit a revised version of the manuscript that addresses the points raised during the review process.

We look forward to receiving your revised manuscript.

Kind regards,

Prof. Anat Gesser-Edelsburg, Ph.D.

Academic Editor

PLOS ONE

Journal Requirements:

"NO"

"The publication cost of his article is funded by Multimedia University, Malaysia."

We note that you have provided funding information that is not currently declared in your Funding Statement. However, funding information should not appear in the Funding section or other areas of your manuscript. We will only publish funding information present in the Funding Statement section of the online submission form. 

"NO"

5. Please include a caption for figure 2.

Reviewers' comments:

Reviewer's Responses to Questions

**Comments to the Author**

1. Is the manuscript technically sound, and do the data support the conclusions?

Reviewer #1: Partly

Reviewer #2: Partly

2. Has the statistical analysis been performed appropriately and rigorously? 

Reviewer #1: Yes

Reviewer #2: Yes

3. Have the authors made all data underlying the findings in their manuscript fully available?

Reviewer #1: Yes

Reviewer #2: Yes

4. Is the manuscript presented in an intelligible fashion and written in standard English?

Reviewer #1: No

Reviewer #2: Yes

5. Review Comments to the Author

Reviewer #1: This paper investigates the role of variables from the Theory of Planned Behavior and message frames (gain-framed vs. loss framed) in vaccination intentions among millennials in South India. I do believe the research is very interesting, relevant, and timely. I was very excited to read the paper based on the preview from the abstract. This research provides an important contribution to the growing literature on health communication during emerging infectious disease. However, despite the overall strength of the topic of this paper and the analyses conducted, the paper itself needs extensive revision across all sections. I’ve provided some commentary to hopefully help in this undertaking.

Strengths of the Paper

1. I appreciated the use of headers in the literature review- they were very helpful for following along

2. It was nice to see the authors report psychometric properties of constructs (assessing for different types of validity and factor loadings). I don’t think this practice is done enough, and it was a pleasant surprise and a particular strength of this manuscript.

3. Excellent job including rationale and description for decisions made about analyses.

Abstract

1. There are a couple of typos-

a. “intention to be vaccination” (line 6)

b. “in both message framing” (line 14)

c. “message framing to user behavioural intentions” (line 20)

2. Authors reference “individual” attitudes on line 7, but then only refer to “attitude” on line 13. I’d keep this consistent, or explain what you mean by individual attitudes- at first I assumed that you investigated multiple different types of attitudes (e.g., cognitive vs. affective or about different health behaviors relevant to COVID-19), but when you refer to “attitude” generally later on when reporting results in the abstract, I wasn’t so sure. It might even be helpful to specify the type of attitude (e.g., attitude toward vaccination?).

Introduction

1. First paragraph: you cite various studies of research conducted in India- were all these studies correlational? I’m just assuming this because you used the term “associated with,” but I want to double check. This sounds like a major gap in the literature. If there is mixed methodology, it would only help to demonstrate that across a range of methodology, people who reported higher vaccine hesitancy also indicated concerns about vaccine safety, rumors, controversies about adverse effects; inadequate knowledge about vaccine benefits; and apprehension based on immunization cost and conflict with traditional cultural beliefs. Generally, I would suggest re-writing this sentence to make it clearer that these are reasons for why people demonstrated vaccine hesitancy about childhood immunization

2. Second paragraph: you reference “level of severity and the extent to which the pandemic is likely to affect the individual” – I’m a bit confused what exactly you are referencing, but it does sound like you are referencing risk perceptions, in which case I would suggest you state and describe this followed by a citation.

3. Second paragraph: “of which was the focus on vaccine hesitancy” is confusing… I’d suggest re-writing this sentence. Was it that one of the four critical areas that they suggested targeting was vaccine hesitancy, or they all were about targeting vaccine hesitancy? Also has there been research to test the efficacy of these four critical areas directly? It’s not clear if the subsequent sentences were about testing these critical areas directly? Also what do you mean by influencers?

4. Second paragraph: you end the last sentence “which could have been avoided.” This came across as somewhat abrupt, and I would recommend removing it and starting a new sentence to get across your point (i.e., it is necessary to test how health communications can be used as part of interventions to target vaccine hesitancy in the context of covid-19)

5. Third paragraph: it would be helpful if you gave examples immediately following your description of loss and gain framed messages.

6. Third paragraph: could you include a transition before bringing up prospect theory? It is unclear to me how one is related to the other, particularly since prospect theory derives from the field of behavioral economics, and Gallagher and Updegraff are psychologists (and the work you cite more largely on persuasive messaging comes from the field of psychology). I see that you describe prospect theory and then bring in an implication based on gain/loss framing and prospect theory, but it doesn’t come across clearly. Examples might be helpful.

7. Fourth paragraph: the previous studies you focus on in this paragraph seem out of place. I’d move them to your literature review, or to your second paragraph where you discuss other studies about covid in India.

Literature Review

1. Do you have rationale for specifically targeting millennials for this research?

2. Theoretical Framework: you present the aim of your study within the literature review on your theoretical framework- it would be helpful to read your study aim earlier.

3. Perceived threat towards COVID-19- I don’t see a citation or explanation for why the rising cases could lead to increased perceived risk or threat. Indeed, participants could instead perceive lower threat or risk if they were engaging in defensiveness, which is a common response to threatening health information.

4. Social norms towards COVID-19 vaccine- you refer to the types of social norms as injunctive and descriptive in this paragraph, but then refer to them as direct/indirect social norms in your abstract. This is confusing.

5. Framing effects and TPB- I would avoid use of language like “it is clear” and rather offer assessment on whether there is strong evidence in support of the theory. In the same section you also refer to an “inability of past literature to produce conclusive results of framing” and so there is a contradiction in how you set this paragraph up.

Hypotheses:

1. You reference a “positive impact” in your hypotheses, but it would be helpful to see written out exactly what that means

2. In H6, what is the significant difference that you hypothesized (if you did hypothesize specific differences a priori)

Materials and Methods

1. Did you pilot test your messages? If not, why not? Also, it’s unclear why you write India’s CDC in parentheses next to the messages. Is it because the information itself was from the CDC? In your text it sounds like the messages were created by the authors of the study. Clarification on whether this text was taken directly from the CDC (perhaps with only slight modification for the gain/loss framing?) or were created based on information from the CDC is helpful to know.

2. You reference “usable responses” for each condition in your study. What were the reasons for exclusion or eligibility?

3. You mention sampling from individuals who are not vaccinated, but are the necessarily vaccine hesitant? Or perhaps they haven’t had an opportunity to be vaccinated yet? Data collection was carried out between August-September 2021- I’d recommend contextualizing when it was that the vaccine was made available and more information about access to the vaccine. I’m seeing later in the limitations section that WHO had not given approval for the vaccines, and so I think this point should be made earlier in regard to your sample… is it still appropriate to characterize the sample as vaccine hesitant if the vaccines were not even approved?

Results

1. I’m seeing many of the analyses/results are described in present tense, rather than past tense.

Discussion

1. You bring up loss aversion almost immediately, but this phenomenon was not brought up earlier in the study- rather, you brought up prospect theory from the field of behavioral economics. How do these ideas relate?

2. You bring up other relevant research, but the discussion is very brief. I would like to see more of a description and integration of the studies you bring up (for example, an important point to consider- were these studies also in the context of COVID-19 vaccination, or vaccination more generally).

Conclusion

1. You say that vaccinations are the “only” tool to curb pandemic spread, and this is not true. Other protective health measures (e.g., hand washing, mask wearing, etc.) are also effective tools to curb pandemic spread. It might be the case that vaccination is one of the strongest tools to curb pandemic spread, in which case you should offer a citation for this point.

Reviewer #2: Review of MS#

Tittle: Message Framing and Covid-19 Vaccine Acceptance among Millennials in South India

In this well written paper the authors aimed to investigate the effect of framing vaccine communication message with gain and loss framing based on the Theory of Planned

Behavior. Following comments may be useful.

- Why authors tried to assess perceived threat. This concept is not brought in the TPB. Please explain it.

- I suggest to remove hypothesis.

- The discussion section need to be extended by well interpretations.

- I also suggest to address to infodemics in the COVID-19 era.

- The following paper may be useful in explain infodemic:

Global Challenge of Health Communication: Infodemia in the Coronavirus Disease (COVID-19) Pandemic.J Educ Community Health. 2020;7(2): 65-67. doi: 10.29252/jech.7.2.65

6. PLOS authors have the option to publish the peer review history of their article (what does this mean?). If published, this will include your full peer review and any attached files.

Reviewer #1: No

Reviewer #2: No

---

## [Author Response · Author response to Decision Letter 0]

3 May 2022

Dear Reviewers, we thank you for providing constructive comments and suggestion to improve the paper. We have address all your comments and made corrections accordingly. The responses are included in the "responses to reviewers" document. Thank you very much.

---

## [Decision Letter · Decision Letter 1]

23 May 2022

Message Framing and Covid-19 Vaccine Acceptance among Millennials in South India

PONE-D-22-05789R1

Dear Dr. Jeyakumar Nathan,

We’re pleased to inform you that your manuscript has been judged scientifically suitable for publication and will be formally accepted for publication once it meets all outstanding technical requirements.

Kind regards,

Prof. Anat Gesser-Edelsburg, Ph.D.

Academic Editor

PLOS ONE

Additional Editor Comments (optional):

Reviewers' comments:

Reviewer's Responses to Questions

**Comments to the Author**

1. If the authors have adequately addressed your comments raised in a previous round of review and you feel that this manuscript is now acceptable for publication, you may indicate that here to bypass the “Comments to the Author” section, enter your conflict of interest statement in the “Confidential to Editor” section, and submit your "Accept" recommendation.

Reviewer #1: All comments have been addressed

Reviewer #2: All comments have been addressed

2. Is the manuscript technically sound, and do the data support the conclusions?

Reviewer #1: Partly

Reviewer #2: Yes

3. Has the statistical analysis been performed appropriately and rigorously? 

Reviewer #1: Yes

Reviewer #2: Yes

4. Have the authors made all data underlying the findings in their manuscript fully available?

Reviewer #1: Yes

Reviewer #2: Yes

5. Is the manuscript presented in an intelligible fashion and written in standard English?

Reviewer #1: Yes

Reviewer #2: Yes

6. Review Comments to the Author

Reviewer #1: I appreciate the authors responsiveness to the lengthy feedback and I thought the authors did a nice job of incorporating the feedback they received into the paper.

Reviewer #2: Authors made all of the corrections based on the reviewers' comments and so I have not additional comments.

7. PLOS authors have the option to publish the peer review history of their article (what does this mean?). If published, this will include your full peer review and any attached files.

Reviewer #1: No

Reviewer #2: No

---

## [Editor Report · Acceptance letter]

30 Jun 2022

PONE-D-22-05789R1 

Message Framing and COVID-19 Vaccine Acceptance among Millennials in South India 

Dear Dr. Jeyakumar Nathan:

I'm pleased to inform you that your manuscript has been deemed suitable for publication in PLOS ONE. Congratulations! Your manuscript is now with our production department. 

Kind regards, 

on behalf of

Prof. Anat Gesser-Edelsburg 

Academic Editor

PLOS ONE